# Bioactive Compounds from *Hermetia Illucens* Larvae as Natural Ingredients for Cosmetic Application

**DOI:** 10.3390/biom10070976

**Published:** 2020-06-29

**Authors:** Cíntia Almeida, Patrícia Rijo, Catarina Rosado

**Affiliations:** CBIOS-Research Center for Biosciences & Health Technologies, Universidade Lusófona, 1749-024 Lisboa, Portugal; cin_almeida@hotmail.com (C.A.); patricia.rijo@ulusofona.pt (P.R.)

**Keywords:** *Hermetia illucens* larvae, biomass composition, bioactives, cosmetic applications

## Abstract

Due to the sustainable organic matter bioconversion process used as substrate for its development, the *Hermetia illucens* (Linnaeus) larvae biomass is considered a source of compounds with high aggregate value and quite a promising market. The materials that can be extracted from *H. illucens* larvae have opened the door to a diverse new field of ingredients, mainly for the feed and food industry, but also with potential applicability in cosmetics. In this review we succinctly describe the larval development and rearing cycle, the main compounds identified from different types of extractions, their bioactivities and focus on possible applications in cosmetic products. A search was made in the databases PubMed, ScienceDirect and Web of Science with the terms ‘*Hermetia illucens*’, ‘bioactives’, ‘biochemical composition’ and ‘cosmetics ingredients’, which included 71 articles published since 1994.

## 1. Introduction

In the last decade a growing demand for natural and renewable sourced ingredients has been noticed, driving different types of industries, such as animal feed, human foods, pharmaceutical and cosmetics to offer the consumer increasingly innovative products.

The sustainable exploitation of natural resources can be further improved if integrated solutions are employed to tackle the problems of overproduction, disposal of solid organic waste and growing demand for natural raw materials. The concept of a closed system of production minimizes residue generation, energetic costs, transportation and greenhouse gas generation [1].

Natural bioactives have been found in nature from diverse sources like plants, animals and different types of microorganisms [2]. Insect mass cultures have become available in recent years as a source for proteins and lipids and are currently considered a promising alternative. There is a sound rationale for this, since insects are highly efficient waste bio converters, induce a low environmental impact, since they emit less climate-damaging greenhouse gases and consume less water, and the risk of zoonosis is reduced [3,4]. Furthermore, they can be fed with organic waste from various sources and can be reared on small surfaces, therefore, making large-scale industrial breeding possible [5].

In this context, the rearing of the black soldier fly (*H. illucens*) larvae has motivated great interest, and applications in several sectors are already envisioned, especially due to their nutritional value (Figure 1).

*H. illucens* larvae are seen as a promising option for providing a sustainable feed source of proteins and lipids [6]. This is mainly linked to their high capacity to convert large quantities of organic waste thanks to the variety of enzymes in their digestive tract [7]. However, this material might also become a potential target for the cosmetic sector, since the *H. illucens* larvae fatty acids profile resembles those of coconut oil or palm kernel oil, and its biomass contains amino acids and enzymes, which may serve as base ingredients for the development of different products [5]. Thus, using a suitable model of production, the black soldier larvae can act as a great biological control and waste management agent [7] and its biomass can provide compounds with high aggregated-value.

The aim of this brief review is to present the biomass composition of *H. illucens* larvae, describe the properties of the compounds obtained from their extracts, as well as identify possible applications in the development of innovative cosmetic formulations.

## 2. *Hermetia Lllucens* Life Cycle and Rearing Process

*Hermetia illucens* (Linnaeus), known as the Black Soldier Fly (BSF) is an insect belonging to the order of Dipterans, family of *Stratiomyidae* [8]. As mentioned previously, this species has a high potential for large-scale production and can be efficiently used to convert low value organic waste into high-value compounds. In order to achieve this, the rearing process must ensure conditions that have satisfactory effects on the growth performance and, thus, composition of *H. illucens* larvae.

In the *H. illucens* life cycle, the transition from the larval to the adult stage occurs following a passage through a nymph phase, each with different morphologies and life habits. This insect species has a short development life cycle lasting 45 days with four stages- egg (4 days), larval (18 days), prepupal (14 days) and adult (9 days) [6,9,10]. The eggs are small (around 1 mm) and present a color range between white and cream, as shown in Figure 2 [1]. The larval stage (with five phases) has morphological variations characterized by a color change from beige to dark brown, and in the prepupal stage a migration out of the substrate occurs [9]. The larvae can thrive in various kinds of decaying organic matter due to their efficient oral structures, rich intestinal microbiota and high enzymatic activity, which allows them to metabolize molecules such as starches, proteins and lipids [8]. The time spent in the larval stage depends on physical conditions and food availability [9]. Their life cycle and nutritional composition can be influenced by the quality and amount of the rearing substrate [7,11]. Temperature is also a key environmental parameter for larval development and survival, enabling variations in the 20–30 °C range [12]. After completing its larval development, the insect enters the prepupal stage, when it lies motionless, while its cuticle is rigidified and becomes rich in salts of calcium, forming a dark envelope. In this stage the larva empties its digestive tract and no longer needs to feed, relying on the nutrients stored during the larval stage [13]. Using the insect in the prepupal stage might offer two advantages: an empty digestive tract, reducing the risk of carrying pathogenic microorganisms, and the prepupal migrating behavior that offers opportunities for harvesting in a large-scale rearing system [14].

*H. illucens* larvae are able to feed on an immense variety of organic material. Different substrates have already been applied in their rearing, ranging from manure, chicken feed, vegetable waste, restaurant waste, as well as winery and brewery agro-industrial by-products [15]. Different studies report that the substrate offered has an impact on the development and biochemical composition of the larvae (Table 1), but concentrations of lipids around 30% and protein around 40% of dry weight (DW) are normally obtained [9].

## 3. Extraction Methods

The extraction process of animal source compounds must be chosen according to the chemical and physical characteristics of the matrix and the product of interest. Given the biochemical composition of insect biomass, extraction methods are based on the solubility of these compounds in aqueous medium or in organic solvents [18].

Many studies have been dedicated to the extraction of the protein, amino acid, and lipid contents from insects. The fat composition range on average 10 to 60% in dry matter between insect’s species and this amount is higher in the larval stages compared to adults [19]. According to the literature, the highest fat content is generally obtained from extraction methods using different types of organic solvents, such as hexane, petroleum ether, methanol and diethyl ether [1,14,18,20,21,22].

Trompa-Souza et al. [18], compared three extraction methods based on water and organic solvents to determine the lipid profile in different insect species. The best yield was achieved using solvents mixture dichloromethane: methanol (Folch method), and the lowest yield was obtained from the aqueous method. In the four insect species studied, the yields using organic solvents-based methods were ranging around 7.5% to 12.9% with the Folch method and 6.0% to 12.7% with the Soxhlet extraction method. Although the yields from aqueous extraction were low (ranging between 1.6% and 7.8%), it should be noted that this method was able to extract omega-3 fatty acids, an important lipid group due to its health benefits. Furthermore, it was observed by these authors that the lipids obtained from methods with organic solvents provide free fatty acids and partial glycerides, which were not extracted by the aqueous method [18]. The presence of these compounds can lead to a future refinement process. Applying the Soxhlet method and using hexane as solvent, Yi et al. [20] achieved a lipid extraction yield ranging between 3.6 % and 16 % from five different species of insects. In order to obtain the lipid fraction present in *H. illucens* larvae, Janssen et al. [21] used petroleum ether in the extraction process and the Soxhlet system and the fat content obtained was between 21 and 24% DW. This result was significantly different from that achieved by Bosch et al. [23], who obtained lower fat contents for *H. illucens* larvae, with a yield around 12.8% of DW, using the same solvent and applying the Soxhlet method. In the assessment of the impact of four types of organic substrate for the *H. illucens* larvae rearing, Spranghers et al. [12] achieved between 336–386 g·kg^−1^ of ether extract from the dry matter, applying the Soxhlet method with diethyl ether. Although the use of organic solvents and the Soxhlet method can result in good yields, it should be noted that the time consumed by this methodology is around 6 h. Following an alternative route for lipidomics analysis in the *H. illucens* larvae, Rabani et al. [24] used water and a biological decomposer. It was observed that when comparing the two experimental groups, control and decomposed, the major fatty acids remained the same, but the oleic acid and linoleic acid content decreased to 18% and 16% respectively, whereas lauric acid and palmitic acid increased to 41% and 16%, respectively [24]. Finally, from the review of the literature it can be perceived that the development of useful and sustainable techniques with use of green solvents for extraction of insect compounds has become an important issue in the chemistry of natural products.

Other studies aimed the evaluation of the protein quality and content, as well as nutritional value and amino acid composition of many insects [25,26]. The protein content of insects is highly variable, and many species contain approximately 60% protein on a dry matter basis [27]. In the quantification of proteins from insect biomass, most of the methods use the nitrogen content, applying nitrogen-to-protein conversion factor (Kp) of 6.25 [20,21,28].. The Dumas method has been applied to the quantification of total nitrogen present in the organic matrix, which is combusted at high temperature in an oxygen atmosphere. This method has been considered a better alternative to the protein and nitrogen quantification due to greater accuracy, speed and productivity [29]. Traditionally, many works use the Kjeldahl method for the determination of the protein content, which measures the total organic nitrogen content based in a distillation and titration process [30]. The amount of protein is obtained from multiplying by a conversion factor. The main limitations of this technique are: (i) slow digestion of samples; (ii) underestimation of nitrogen contents; (iii) use of dangerous reagents at high concentrations; (iv) significant volumes of waste requiring neutralization for discarding [31].

The crude protein contents might be higher than the real protein content, since significant amounts of nitrogen are also bound to the exoskeletons as chitin. The techniques related with protein extraction are based on the use of previously frozen or N2-frozen treated larvae. For extracting and obtaining protein fractions of different insect species, Yi et al. [20] applied an ascorbic acid solution, followed by homogenization, filtration and centrifugation processes to separate the fractions. For quantification of the protein, the authors applied the Dumas method and the results showed that the crude proteins present in the insect species ranged from 19–22%, including the chitin nitrogen. A contrasting technique was applied by Janssen et al. [21], blending the frozen larvae in citric acid solution with 0.2 M disodium phosphate buffer. The obtained solutions were centrifuged and the supernatant was filtered and dialyzed at 4 °C. The protein content (quantified by the Dumas method) ranged from 36 to 49% within larvae of the three insect species studied, from the calculated mean value of Kp 4.76 ± 0.09. The protein content estimated in *H. illucens* larvae is 36% based on dry weight [21]. This value is consistent with that obtained by Diener et al. [9], who obtained results ranging 31.9% to 46.3% of their dry matter (based on the Dumas method) in the estimate of the protein content in larvae. However, higher protein contents, ranging between 37 and 42%, have been reported in the literature [22,32].

In summary, the utilization of materials from BSF larvae or other insects as ingredients for food, feed, pharmaceuticals or cosmetics requires not only their mass production, but also viable, effective, well-yielding and industrially applicable methodologies for obtaining compounds of interest from insect biomass.

## 4. *H. illucens* Larvae Bioactives Composition and Cosmetic Applications

Currently, much attention has been directed to research of natural sources of biologically active substances to be used in cosmetic formulations [33]. Increasingly, the cosmetic industry has been challenged to use more natural and sustainable raw materials, since environmental issues have become one of the main factors that determine the growth of this type of products [34]. Moreover, the consumers are increasingly concerned about the safety of the cosmetic formulations, as well as with the composition and origin of the raw materials that make up these products.

As previously stated, insect production is a potential market business because it is possible to obtain inexpensive and bioactive high value compounds from their biomass [35]. Though the idea of cosmetic products based in insects would be repulsive to many consumers, it is reasonable to consider that extracting insect materials and using these as cosmetic ingredients may increase consumer acceptance [18]. As previously explained, the content of compounds obtained from insect biomass varies considerably, and depends on the stage of metamorphosis, origin of the insect and its diet [36]. Apart from *H. illucens*, studies addressing the potential use in skin care of different types of insects have explored locusts, house crickets [5] and silkworms [37]. As mentioned above, the *H. illucens* larvae biomass is so rich that depend on their fat content it can even provide sufficient energy for animal or human diet, as well as satisfactory amounts of proteins and amino acid requirements [6,38]. It is also important to highlight the high contents of mono and polyunsaturated fatty acids, polysaccharides and chitin. Furthermore, it contains micronutrients such as copper, iron, magnesium, manganese, phosphorus, selenium and zinc, as well as vitamins [6,39].

In the next sections, the main compounds obtained from *H. illucens* larvae biomass and their possible applications as cosmetic ingredients will be highlighted.

### 4.1. Proteins and Peptides

In general, proteins and peptides are used in cosmetic formulations due to their important role in skin homeostasis and because they can help to maintain skin hydration and to decrease the rate of skin aging [40,41]. Collagen and collagen peptides, mainly extracted from animal sources, are two examples of a biomaterials with these cosmetic applications [42].

According to different studies focused on *H. illucens* larvae as a source of protein for animal feed, this species has a high protein content, regardless the type of substrate that is used in its rearing [14]. However, from the data collected in the literature it can be seen that there are considerable differences amongst the amino acids profiles of larvae reared on different substrates (Table 2). Taking the particularities of each extraction and quantification method into consideration, it can be observed that aspartic acid, glutamic acid, leucine and valine are the predominant amino acids present in *H. illucens* larvae, regardless of the substrate. Despite not being the most abundant, arginine and glycine are also present in considerable amounts, and these amino acids have a high applicability as cosmetic ingredients. Glycine is naturally present in the skin as part of the Natural Moisturizing Factor (NMF) [43] and it can also be used as a buffering agent. It has been reported that arginine has important holes: protects the skin from free radicals, acts as an antioxidant, boosts collagen production, in cell division and in the healing process [44]. Arginine can also act as a humectant, increasing the *stratum corneum* hydration. Proline, also present in considerable quantities in some of the mentioned substrates, is an amino acid present in the collagen molecule and which is part of the NMF, acting as a moisturizer [44].

It is well known that insects have a well-developed innate immune system, subdivided into cellular and humoral defense responses [45]. Survival in decomposition habitats requires substances possessing antibiotic activity as a powerful defense against microorganisms. Antimicrobial peptides (AMPs) are natural antibiotics, which have the ability to kill, or inhibit the growth of various microorganisms that mainly attack the cell envelope [47]. Within the potential bioactive substances that can be extracted from the *H. illucens* larvae, their capacity for high expression of AMPs and other substances possessing activity against drug resistant “superbugs” can also be considered [45,48]. A study on the antibacterial activity against methicillin resistant *Staphylococcus aureus* (MRSA) suggests that more than one “antibacterial” substances act in synergy to increase the effect [49].

Recent studies reported the antimicrobial activity of a *H. illucens* larval extract on several Gram-positive and Gram-negative microorganisms. Amongst the classes of AMPs present, Müller et al. [45] found high expression of defensins, cecropins, attacins and diptericins. Furthermore, these authors reported that: (i) the majority of insect defensins act strongly against Gram-positive microorganisms, e.g., *Bacillus subtilis* or *Staphylococcus aureus*; (ii) cecropin peptides exhibit a broad antimicrobial activity against Gram-positive and Gram-negative microorganisms as well as against fungi; (iii) attacins are antimicrobial active against *Escherichia coli* and some other Gram-negative bacteria, e.g., *Acinetobacter calcoaceticus* and *Pseudomonas maltophilia*; (iv) diptericins inhibit the growth of the Gram-negative bacteria *E. coli* and *Salmonella typhimurium.* Defensins and cathelicidins are currently some of the most widely studied classes of AMPs in the human skin [47].

In addition, AMPs have been found to have potential as antioxidants, anti-inflammatory agents and promote keratinocyte proliferation [47], and the antioxidant activity in relation to photoinduced lipid peroxidation of melanin and ommochromes isolated from *H. illucens* was determined [50]. However, it is important to stress that a high production of AMPs by the larvae is caused by a pre-infection with pathogenic organisms (to induce the immune response) or through substrates offered for feeding with a high pathogenic load.

In this context, AMPs obtained from *H. illucens* larvae can be considered substances with interesting biochemical, scientific and commercial properties and this makes them attractive candidates for the development of novel antimicrobials [49]. These could be used as therapeutic ingredients for the treatment of cutaneous diseases related with the skin microbiota and to promote skin health. Additionally, they could be employed as alternative preservatives in cosmetics, since the safety of conventional preservatives (such as parabens) is currently being questioned by consumers. They could be incorporated in topical formulations such as lotions, creams, shampoos, and could therefore be valuable ingredients for the cosmetic industry [51].

### 4.2. Lipids and Fatty Acids

Lipids are also a main component of insects biomass and, as such, can be a considerable source of fat [20]. Insect oils are rich in saturated and polyunsaturated fatty acids, in addition to commonly found essential fatty acids such as linoleic and α-linolenic acids [52]. The insects lipid composition is also variable according to their species and life stage [18]. In the case of *H. illucens*, during larval development their lipid content increases from about 13% to 40% before pupation. Furthermore, predictably, the storage of lipids by *H. illucens* larvae will depend of its dose in feeding, meaning that the higher the lipid content in the growth substrate, the higher the lipid content of larvae [8]. The oil extracted from *H. illucens* larvae has a high quality and it is consider a byproduct of animal feed production [45].

Based on a comparison of recent studies, it is possible to observe the main fatty acids profile in *H. illucens* larvae (Figure 3). Undoubtedly, the most abundant is lauric acid representing 36–60% of total fatty acids, followed by palmitic, oleic, myristic and linoleic acids.

The high content of lauric acid in larvae oil is similar to that of coconut oil, and it should also be noted that many of the beneficial properties of this oil have been attributed to this component. A 12-carbon chain fatty acid, lauric acid is biologically active and has been shown to have potent antibacterial and antiviral activities [56]. Medium chain fatty acids have been reported to have the ability to treat severe bacterial infections that are antibiotic resistant [56]. Lauric acid can be converted into monolaurin, which is known to display antiviral, antibacterial, and antiprotozoal activities [54]. Thus, larvae oil seems promising as a possible alternative preservative. In addition, lauric acid and other fatty acids present in the *H. illucens* larvae lipid fraction can act as emulsifiers and stabilizers of disperse systems, and are commonly used in the production of soaps and cosmetics [54,56]. Verheyen et al. [5] suggest that due to the high concentration of lauric acid in *H. illucens* oil, its application in skin cosmetic formulations could be enhanced to that of other oils of natural origin. The authors also recommended the need for refining to remove phospholipids and unsuitable free fatty acids, as well as to improve its color and odor characteristics.

As mentioned above, fatty acids are frequently used in cosmetics, thus other components of *H. illucens* larvae oil can be applied, either isolated or as a lipidic structural complex [57]. Palmitic, oleic and linoleic acid, along with other essential fatty acids have been identified as key components in the human epidermis [58]. Oleic acid activates the lipidic metabolism, restoring the skin barrier and enabling the retention of moisture in the *stratum corneum* [59]. Myristic acid is used in cosmetic formulations for thickening and stabilizing emulsions, but it can also restore the cutaneous barrier properties and enhance the permeability of active components into the skin [59]. Palmitic acid and its derivatives are used in creams and lotions as emulsifiers and emollients. Linoleic acid has a physiological role in maintaining the water permeability barrier of the skin, since it is a constituent of acylglycosyl ceramides, which play a structural role in the *stratum corneum* [60]. It has been suggested that linoleic acid deficiency can result in development of dermatitis in adults and children.

Although not highlighted in the graph in Figure 3, omega-6 fatty acids (ω-6 PUFAs) were also identified in the lipid composition of *H. illucens* biomass. ω-6 PUFAs have shown beneficial effects in cardiovascular diseases and deficit of these essential fatty acids promotes skin inflammation and impairs wound healing [61].

As described above, fatty acids provide an invaluable contribution to the skin barrier. Thus, an increase in the application of fatty acids as formulation ingredients has been noted, not only due to their functional effects (biological activities), but also for their high biocompatibility.

### 4.3. Polysaccharides

The exoskeleton of insects is structurally composed of chitin, a nitrogen-containing polysaccharide present in the cuticle of *H. illucens* larvae [8]. After cellulose, chitin is the most abundant natural polysaccharide on Earth [62]. Chitin is an insoluble fiber, a non-toxic and biodegradable linear polymer. It has been considered a new functional biomaterial with potential application in distinct fields. Despite its low solubility, chitin is considered a good material for fibers feedstock. Chitin fibers can be used as absorbable sutures and wound treatment materials, and have been reported to be able to accelerate the rate of wound healing by about 75% [63]. Additionally, recent studies showed that chitin has effects on the innate and adaptive immune-response, including the ability to recruit and activate immune cells and induce cytokines [64]. Chitin is also associated with the defense mechanisms against some parasitic infections and could reduce allergic reactions to certain entities [65].

After a deacetylation process chitin is transformed into chitosan [66]. This polymer has many applications and is commonly used in water purification and wastewater treatment. But its use in other industrial sectors is growing, from agriculture to pharmaceutical, cosmetics and textile. The properties of chitosan are dependent on many structural factors that can be tailored. Molecular weight, the degree of deacetylation, pH and other features have been shown to play an important role in the final biological activity of chitosan [67]. In health products, chitosan is already used as a carrier or a vector for drug controlled release and in vaccines, but there are also reports of its use against tumors, obesity and high cholesterol [65]. As a cosmetic ingredient, chitosan can work as a hydrocolloid, having the peculiarity of being the only cationic gum of natural origin that becomes viscous after neutralization with acids [68]. This property facilitates its interaction with the skin and hair. Chitosan has also the advantage of being compatible with other biologically active components incorporated in cosmetic formulations [62].

Chitin, chitosan and their derivatives have high application in hair care, and are currently used in shampoos, conditioners, hair tonics, among other products. These molecules and hair have opposite electrical charges [68] forming a clear, elastic film, improving hair softness, smoothness, and elasticity. In skin care products, chitosan also has advantages due to its positive electrical charge and high molecular weight, enabling retention of water molecules by hydrogen bonding [62]. These proprieties prevent skin penetration, allowing chitosan to form a superficial film and act as a moisturizing agent in creams, lipsticks and lotions.

In addition to the previously mentioned activities, the ability of chitin, chitosan and their derivatives to eliminate free radicals via scavenging activity in in vitro and in vivo assays has been reported [66,67]. This antioxidant activity can be used in products aiming to protect the skin against damage by oxidative stress, or to prevent the oxidation of other active ingredients in cosmetic formulations, such as essential oils or vitamins [34].

Chitosan has shown an inhibitory effect against bacteria and fungi, thus having applicability as preservative [69,70]. It is suggested that the antibacterial activity could occur via changes in the bacterial membrane permeability, breakdown of the cytoplasmic barrier or the blockage of nutrient transport, resulting in cell lysis. The degree of deacetylation is a factor that enhanced the antibacterial activity, that is, amino groups seem to play a functional and essential role [71]. Chitosan and some derivatives (mainly cationic derivatives) exhibited antimicrobial activity against Gram-positive and Gram-negative bacteria [69,71]. The mechanism of action in bacteria as E. coli is via an electrostatic interaction with the anionic bacterial surface. The binding of lower molecular weight chitosan derivatives with DNA or RNA inhibited Gram-positive bacteria like S. aureus [49].

## 5. Conclusions

*H. illucens* larvae extracts are a promising source of bioactive substances and can play an important role in the development of innovative cosmetics. Due to the sustainability of its rearing, mass cultures of these insects seem to be economically viable. The larvae oil and the protein and chitin content have vast cosmetic applications, and these substances are worth considering as high value co-products. The main challenges associated with their use will be linked with the development of methods to isolate the extracted materials on an industrial scale.

## Figures and Tables

**Figure 1 biomolecules-10-00976-f001:**
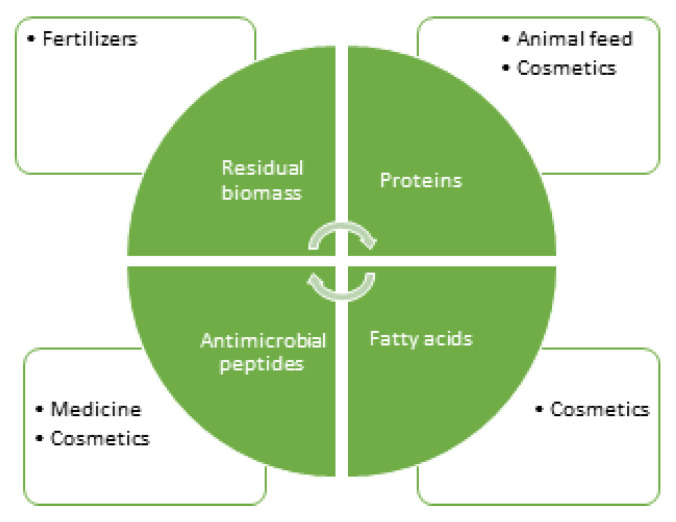
Flowchart of materials obtained from *Hermitia illucens* larvae and potential industrial applications.

**Figure 2 biomolecules-10-00976-f002:**
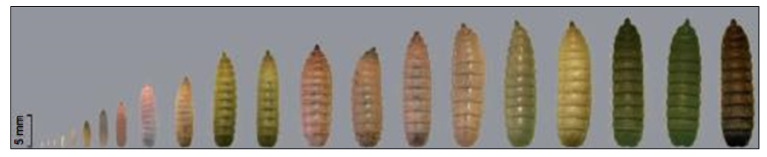
Development process of H. illucens, from eggs to the larvae stage. Source: Adapted from Caruso et al. [8].

**Figure 3 biomolecules-10-00976-f003:**
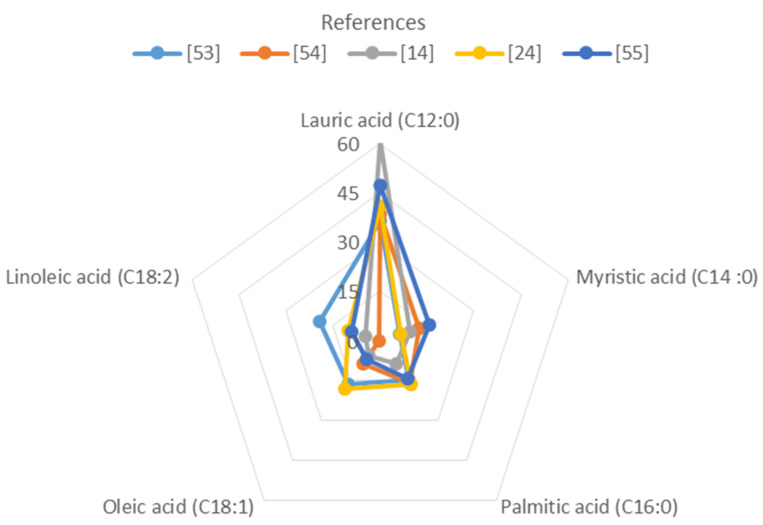
Composition of main fatty acids in the *H. illucens* larvae (percentage of oil content) [14,24,53,54,55].

**Table 1 biomolecules-10-00976-t001:** Comparison of the biochemical composition of *H. illucens* larvae reared on different substrates.

Substrate Type	Larvae Composition			References
	Crude Protein	Lipids	Chitin	
1. Chicken feed	412 (0.6) ^a^	336 (0.4) ^a^	62 (2.8) ^a^	[14]
2. Vegetable waste	399 (0.2) ^a^	371 (1.1) ^a^	57 (1.8) ^a^	
3. Restaurant waste	431 (0.6) ^a^	386 (2.3) ^a^	67 (1.3) ^a^	
4. Biogas digestate	422 (1.4) ^a^	218 (0.5) ^a^	56 (1.5) ^a^	
5. Poultry feed	39.6 ± 0.2 ^b^	-	-	[16]
6. Food waste	39.2 ± 2.5 ^b^	-	-	
7. Fruits and vegetables	41.3 ± 1.0 ^b^	-	-	
8. Poultry manure	41.6 ± 1.5 ^b^	-	-	
9. Fruit waste	307.5 ± 10.2 ^d^	10.04 ^c^	56.0 ± 3.9 ^d^	[15]
10. Winery by-product	344.3 ± 7.6 ^d^	73.57 ^c^	52.9 ± 9.2 ^d^	
11. Brewery by-product	529.6 ± 5.2 ^d^	82.47 ^c^	14.2 ± 6.0 ^d^	
13. Seaweed (*Ascophyllum nodosum*)	41.3 ± 1.1 ^e^	22.2 ± 0.2 ^f^	-	[17]

^a^ Means (and coefficients of variation) in g·kg^−1^ dry matter; ^b^ percentage of dry matter: N-total × 6.25; ^c^ g·kg^−1^ dry matter; ^d^ obtained using the nitrogen-to-protein conversion factor of 6.25; ^e^ percentage, dry weight and nitrogen × 6.25; ^f^ percentage, dry weight.

**Table 2 biomolecules-10-00976-t002:** Comparison of the amino acid composition of *H. illucens* larvae reared on different substrates.

Amino acids	Substrates				
	Chicken Feed ^a^ [14]	Organic Waste ^b^ [45]	Cattle Manure ^c^ [21]	Human Faeces ^d^ [16]	Undefined ^e^ [46]
Alanine	25.2	6.2	3.7	65.3	12.2
Arginine	20.3	6.2	2.2	51.0	12.3
Aspartic acid	**37.8**	**10.3**	**4.6**	**90.0**	**16.5**
Cysteine	2.5	0.5	0.1	6.7	1.02
Glutamic acid	**41.9**	**12.2**	**3.8**	**105.9**	**19.7**
Glycine	22.6	5.4	2.9	59.2	9.14
Histidine	13.6	4.8	1.9	33.0	5.94
Isoleucine	17.2	4.8	2.0	46.7	7.62
Leucine	**28.6**	**7.7**	**3.5**	**70.0**	**12.1**
Lysine	23.4	7.4	3.4	61.7	11.9
Methionine	7.6	0.6	0.9	18.5	3.37
Phenylalanine	17	6.2	2.2	44.2	7.56
Proline	22.5	6.2	3.3	57.9	10.2
Serine	16.6	4.1	0.1	43.9	7.02
Threonine	16.4	4.5	0.6	37.6	6.82
Tryptophan	Not analyzed	Not analyzed	0.2	16.9	3.00
Tyrosine	6.7	6.0	2.5	**108.8**	12.1
Valine	**24.1**	**6.7**	**3.4**	**65.9**	**12.9**

^a^ g·kg^−1^ dry matter; ^b^ % dry matter; ^c^ g·100g^−1^ Protein; ^d^ g·kg^−1^ crude protein; and ^e^ g·kg^−1^ frozen matter.

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
