# Peer review of "Bioactive Compounds from *Hermetia Illucens* Larvae as Natural Ingredients for Cosmetic Application"

_biomolecules, 2020, doi:10.3390/biom10070976_

Round 1

Reviewer 1 Report

The Authors this review described the Hermetia illucens larvae development and rearing cycle, the main compounds identified from different types of extractions, their bioactivity and focused on possible applications in cosmetic products. The review-paper contains good and novel data. The Authors investigated an interesting and novel topic, and the objective of the review-paper is of worldwide interest and fits well within the overall scope of the journal and special issue. The available literature findings have been accurately discussed and compared.

However, the manuscript could be further improved: firstly after an overall check of the English language. Moreover, the review-paper could be improved by referring to very recently published papers (last 2 years) investigating similar theme. Lastly, the description of Tables can be also further improved.

So, based on my opinion the manuscript merits the acceptance in Biomolecules after revision.

Author Response

We would like to acknowledge the referee for these pleasant comments. 

The manuscript has been thoroughly checked for English language, and several corrections were made. Recently published papers investigating similar themes have been added. The description and legends of the Tables have been improved.

Reviewer 2 Report

Major observations.

The description of properties from the extracts of H. illucens is superficial.

The possible applications are neither described nor detailed.

The formulations of cosmetics are not indicated and, consequently, the innovative formulation neither.

General observations.

The authors describe this review as brief, but may be improved giving more detailed information in many sections. Even, general information regarding cosmetic industry may be incorporated as well as the use of other insect-derived products in this field.

Indicate duration of life cycle and each stage.

It is correct writing “larvae life cycle” if larvae is a stage of life cycle”?

Methods cited in extraction methods section were applied only to larvae stage?

Provides a more detailed information about content of bioactive compounds, metamorphosis stages, energy supply and aminoacid requirements as well as fatty acids, sugars and mineral content (lines 167 to 174).

Cite examples of these bioactives as well as its effects.

Line 178. Mention the origin source for the proteins and peptides used for cosmetic elaboration.

Provide examples of these molecules.

If it is possible, provide a table with the protein composition of H. illucens (similar to Table 2).

Also is necessary providing detailed information about compounds that confers drug resistant since are mentioned only superficially.

Regarding to antioxidant and anti-inflammatory properties of compounds isolated from H. illucens, the bibliographical information is insufficient.

Section “4.2. Polysaccharides” (Correct the numeration to 4.3)

Correct the sentence “graph in Figure 2” since Figure 2 refers to development of H. illucens.

Information about elaboration and composition cosmetics, and examples of them are not provided, which is necessary to justify the use of H. illucens-derived compounds.

Author Response

Please find below a point-by-point response.

Major observations.

The description of properties from the extracts of H. illucens is superficial.

The possible applications are neither described nor detailed.

The formulations of cosmetics are not indicated and, consequently, the innovative formulation neither.

We would like to thank the referee for these comments. The review aims to provide a brief, and not exhaustive, overview of the compounds that are present in the extracts of H. illucens and present possible applications for such materials. We find unfair the comment that the possible applications are not described nor detailed and that cosmetic formulations are not indicated, since sections 4.1, 4.2 and 4.3 are entirely dedicated to this. It is not an objective of the paper to address a specific innovative formulation.

General observations.

The authors describe this review as brief, but may be improved giving more detailed information in many sections. Even, general information regarding cosmetic industry may be incorporated as well as the use of other insect-derived products in this field.

Recently published papers investigating similar themes have been added. Given the objectives of the review, and since it already includes general information on the current trends in the cosmetic industry, such as the interest in ingredients of natural origin or the search for alternative preservatives, a sentence describing studies using insect derived products has been added in line 290.

Indicate duration of life cycle and each stage.

The duration of life cycle and each stage was added from line 86-89.

It is correct writing “larvae life cycle” if larvae is a stage of life cycle”?

In compliance with this comment, the title of section 2 was changed to “H. illucens life cycle”.

Methods cited in extraction methods section were applied only to larvae stage?

Yes, the majority of the studies cited are applied to H. illucens in the larvae stage, since this is the phase of the cycle with higher applicability potential due to the richness of the biomass.

Provides a more detailed information about content of bioactive compounds, metamorphosis stages, energy supply and aminoacid requirements as well as fatty acids, sugars and mineral content (lines 167 to 174).

Given the objectives of the review, we find that these issues have already been addressed in tables 1 and 2. Additionally, it is not possible to find all the information requested in the studies found in the literature.

Cite examples of these bioactives as well as its effects.

We find this comment difficult to understand and kindly ask the reviewer to be more specific on where these examples should be inserted, since throughout sections 4.1, 4.2 and 4.3 the bioactivity of several compounds is described.

Line 178. Mention the origin source for the proteins and peptides used for cosmetic elaboration.

Provide examples of these molecules.

A reference to the use of collagen and its peptides in cosmetics has been added in line 290.

If it is possible, provide a table with the protein composition of H. illucens (similar to Table 2).

This suggestion would certainly enrich the review, but unfortunately we have not found sufficient data in the literature to comply with this request.

Also is necessary providing detailed information about compounds that confers drug resistant since are mentioned only superficially.

The studies found to date have been unable to identify the specific AMP that is responsible for the antimicrobial action against drug resistant bacteria such as SARS, and suggest that is probably the presence of more than one “antibacterial” substance acting in synergy. This information has been added in line 457.

Regarding to antioxidant and anti-inflammatory properties of compounds isolated from H. illucens, the bibliographical information is insufficient.

The work of Ushakova et. al has been added (reference 50).

Section “4.2. Polysaccharides” (Correct the numeration to 4.3)

This correction has been made

Correct the sentence “graph in Figure 2” since Figure 2 refers to development of H. illucens.

This correction has been made since, indeed, the sentence refers to figure 3 (line 492).

Information about elaboration and composition cosmetics, and examples of them are not provided, which is necessary to justify the use of H. illucens-derived compounds.

Given the objectives of this review and the corrections that have been made to the manuscript, we find that sufficient information linking the bioactivity of the compounds and the types of cosmetic formulations where these can be applied has already been supplied.

Reviewer 3 Report

The review paper covers the recent research on Hermetia illucens (Linnaeus) larvae biomass use as new high value ingredients source. In particular, the possible use of insect derivatives in cosmetics, beside the already well-documented applications for the feed and food industry, are described in this specific paper.

Essentially all commercially valuable available component with a wide description of extraction methods and possible bioactivities are carefully described and referenced. The paper is interesting and could be a reference tool for the researchers approaching the field.

An increased appeal for the reader could be introduced presenting also (where available) a comparisons of costs, with respect to the classical sources, for some of the considered larvae biomass derived compounds. This just to support also the conclusion that “mass cultures of these insects seem to be economically viable”. Reporting also in the text some chemical structures for less obvious compounds such as a AMPs, for example, could result in an even better outcome, of an already good work, for publication in "biomolecules".

Author Response

We would like to acknowledge the referee for these pleasant comments.

An extensive revision of the manuscript has been performed and its content has been signficantly improved. No information was found regarding a comparison of the costs of mass production of insects, and the conclusion that “mass cultures of these insects seem to be economically viable” is based in references 3 and 4. Taking into consideration the objectives of this review, we find that adding chemical structures, for example of AMP’s, would not be essential.

Reviewer 4 Report

THe authors submitted a review on bioactive compounds from H. illucens larvae as natural ingredients for cosmetic application. 

The reviewer has the following comments:

  1. English language and style have to be corrected (for example, line 143 and may others).
  2. Review is under referenced: For example lines 29-31, 118-120, 130, 126-134 and many others.
  3. Reference 13 is not the same as in the ref list (line 94)
  4. Description of extraction methods is very confusing. Folch method uses 2:1 chloroform:methanol and lipids are concetnrated in chloroform layer, not methanol (methanol is too polar for that). It is not clear which methods are "aqueous" and which are "organic". For example, Folch method uses both types of solvents. 
  5. It is not clear what is Kp and what is he gaining of 6.25 value (line 126).
  6. Although it is mentioned that chitin and chitosan can eliminate free radicals and possess antimicrobial activity, no details on possible mechanistic aspects of these bio activities was disclosed. 

Major points. 

  1. Analysis of the literature is not completed. For example, it should be interesting to analyze if composition data presented in this review have been affected by a stage of larvae. To do so, the authors would need to include stage of larvae in the tables 1 and 2. 
  2. Arginine and glycine were noted as the most important amino acids that can be utilized in cosmetic industry, but no discussion on how they can be extracted and purified from larvae  and if there are any methods to do so on industrial scale was made.
  3. Lipid extraction was discussed, but nothing was mentioned about purification of the extracted material ( and feasibility of this process on the industrial scale). 

Author Response

Please find below a point-by-point response to the review:

  1. English language and style have to be corrected (for example, line 143 and may others).

The manuscript has been thoroughly checked for English language, and several corrections were made.

2. Review is under referenced: For example lines 29-31, 118-120, 130, 126-134 and many others.

New references have been added to the manuscript, and ref 2, 29, 30 and 33 have been added to the examples indicated by the reviewer.

3. Reference 13 is not the same as in the ref list (line 94)

This mistake has been corrected, since the reference we intended to cite was that of Trompa-Souza et al. 2014

4. Description of extraction methods is very confusing. Folch method uses 2:1 chloroform:methanol and lipids are concetnrated in chloroform layer, not methanol (methanol is too polar for that). It is not clear which methods are "aqueous" and which are "organic". For example, Folch method uses both types of solvents. 

The correction was made for the term "solvent mixture dichloromethane: methanol". In the mentioned article, the aqueous method is that in which water is used as an extractor solvent. The methods that use organic solvents (even if it is from a mixture of solvents with different polarities) were compared with one that used only water.

5. It is not clear what is Kp and what is he gaining of 6.25 value (line 126).

Nitrogen-to-Protein Conversion Factors (Kp), generally used for determination of protein content. This information was added in line 229.

6. Although it is mentioned that chitin and chitosan can eliminate free radicals and possess antimicrobial activity, no details on possible mechanistic aspects of these bio activities was disclosed.

The antimicrobial activity of chitosan is described in detail in the paragraph starting at line 502. The free-radical scavenging activity of these compounds has been added in line 499.

Major points. 

  1. Analysis of the literature is not completed. For example, it should be interesting to analyze if composition data presented in this review have been affected by a stage of larvae. To do so, the authors would need to include stage of larvae in the tables 1 and 2. 

From the information obtained from the present review, it can be seen that the composition of the larvae is strongly associated with the type of substrate on which they are raised, since the larval stages are those where storage of energy and of nutrients occurs.

2. Arginine and glycine were noted as the most important amino acids that can be utilized in cosmetic industry, but no discussion on how they can be extracted and purified from larvae  and if there are any methods to do so on industrial scale was made.

We agree with the reviewer’s comments, however, the objective was to highlight those amino acids present in biomass that have greater application and cosmetic importance.

3. Lipid extraction was discussed, but nothing was mentioned about purification of the extracted material ( and feasibility of this process on the industrial scale). This comment is similar to that raised in point 2. A sentence addressing these issuee was added to the conclusion section.

Round 2

Reviewer 2 Report

In the previous communication, the authors were asked for some questions and observations that have been kindly provided in the most recent version.

After this review I consider that actual manuscript is more complete and precise than the previous one.

Reviewer 4 Report

I recommend to accept this manuscript as it is.